# The Sweet Side of Constipation: Colonic Motor Dysfunction in Diabetes Mellitus

**DOI:** 10.3390/nu17193038

**Published:** 2025-09-24

**Authors:** Michelantonio De Fano, Sara Baluganti, Marcello Manco, Francesca Porcellati, Carmine G. Fanelli, Gabrio Bassotti

**Affiliations:** 1Endocrine and Metabolic Sciences Section, Department of Medicine and Surgery, University of Perugia, 06132 Perugia, Italy; marcellomanco.mm@gmail.com (M.M.); f.porcellati.fp@gmail.com (F.P.); carmine.fanelli@unipg.it (C.G.F.); 2Gastroenterology, Hepatology and Digestive Endoscopy Section, Department of Medicine and Surgery, University of Perugia, 06132 Perugia, Italy; sara.baluganti@gmail.com (S.B.); gabassot@tin.it (G.B.)

**Keywords:** constipation, diabetes mellitus, GLP-1 RAs, laxatives

## Abstract

**Background/Objectives**: Chronic constipation is a prevalent gastrointestinal (GI) disorder among individuals with diabetes mellitus (DM), occurring more often than in healthy subjects. This review provides a systematic overview of this often-underestimated clinical condition in people with DM. **Methods**: A narrative review of literature up to 30 May 2025 was conducted, focusing on studies regarding the pathogenesis of constipation in DM, the correlation with GLP-1 RAs treatment, and the diagnostic-therapeutic framework. **Results:** The mechanisms underlying constipation in DM remain largely unclear; however, a multifactorial etiology has been proposed, involving structural changes in various tissues within the GI tract wall, as well as functional abnormalities, often secondary to hyperglycemia. It is noteworthy that the use of GLP-1 RAs, a class of medications crucial for managing glycemic control and reducing cardiovascular and renal risk in type 2 DM, is another cause of constipation. The diagnosis of constipation is typically based on clinical evaluation, as validated methods for assessing colonic transit are invasive and available only in specialized centers. Treatment objectives include alleviating symptoms and restoring bowel function. The primary strategy for management involves dietary changes and physical activity. If the clinical response is inadequate, the use of laxatives is recommended. Finally, newer agents and mechanical methods may be considered for scenarios that are particularly severe. **Conclusions**: Given the increasing global prevalence of DM, healthcare professionals must recognize the clinical problem constituted by the occurrence of chronic constipation, especially considering the use of medications such as GLP-1 RAs that may induce this clinical condition.

## 1. Introduction

Diabetes mellitus (DM) is a chronic disease characterized by elevated blood sugar levels and varying degrees of insulin deficiency and/or resistance. The major types of DM are type 1 (T1DM) and type 2 (T2DM), with T2DM being significantly more prevalent, accounting for over 90% of adult cases [1].

Specialized cells known as glucosensors, distributed throughout the gastrointestinal (GI) tract, are able to detect fluctuations in glucose levels and, in response, transmit hormonal/neuronal signals to regulate glycemia in tissues involved in glucose metabolism [2]. Consequently, the frequent impact of DM on the GI tract [3], particularly in the presence of autonomic neuropathy (diabetic autonomic neuropathy, DAN), is of clinical significance. The cumulative effect of DM on the GI tract is termed “Diabetic gastroenteropathy” [4].

Although DM involvement, particularly concerning sensorimotor functions, may occur from the esophagus to the anorectum [5], the most well-documented, frequent, and significant abnormalities have been observed in the stomach [6,7]. However, it is important to note that the large bowel is also frequently affected [8].

Among the various GI disturbances, constipation is particularly frequent in individuals with DM [8]. According to the Rome Criteria IV, constipation is defined by the presence of two or more of the following: (a) straining during more than ¼ (25%) of defecations; (b) lumpy or hard stools (Bristol Stool Form Scale 1–2) in more than ¼ (25%) of defecations; (c) sensation of incomplete evacuation in more than ¼ (25%) of defecations; (d) sensation of anorectal obstruction/blockage in more than ¼ (25%) of defecations; (e) fewer than three spontaneous bowel movement (SBM) per week; (f) loose stools rarely present without the use of laxatives; and (g) insufficient criteria for irritable bowel syndrome [9]. These criteria should be met for the last three months, with onset at least six months before diagnosis.

According to a previous systematic investigation, the prevalence of constipation in the general population ranged from 0.7% to 79%, with a median of 16% [10], with a greater rate in women (10.2% vs. 4% in men) [11]. It is even more prevalent among older adults and those with lower socioeconomic status [12]. The prevalence of constipation is greater in individuals with DM compared to the general population: in a study on 136 subjects [13], almost 60% had constipation symptoms, whereas other studies have found a prevalence ranging from 11 to 56% [14,15].

Constipation is associated not only with a diminished quality of life (QoL) [16] but also with an elevated risk of cardiovascular disease (CVD) and premature mortality [17,18]. Consequently, given the significant increase in the prevalence of DM and its subsequent impact on healthcare costs, constipation should be included among the conditions warranting attention and potential treatment in routine clinical practice (Figure 1).

## 2. Pathophysiology of Constipation in Diabetes Mellitus

The mechanisms underlying constipation in DM have not been carefully investigated and remain poorly understood. However, known observations suggest that the etiology is likely multifactorial (Figure 2).

Similar to idiopathic chronic constipation, both colonic dysmotility and anorectal dysfunctions (i.e., impaired anal sphincteric relaxation during defecation) may contribute to constipation in DM [19]. Additionally, DM induces alterations in gut structure and biomechanical properties, which form the basis for fundamental changes in sensory-motor functions [20]. In individuals with DM, these sensory-motor changes in the gut may reflect structural and functional alterations in the peripheral nerve, enteric nervous system (ENS), and central nervous system (CNS). Gut sensory-motor disorders appear to be more prevalent in cases of more severe neuropathy [21].

More than 30% of long-standing subjects with DM exhibit DAN, making it one of the most prevalent complications of DM, and it is correlated with other GI complications [22]. Peripheral DAN can affect the ENS and sensory nerves [23,24]. Evidence suggests that DAN-related alterations occur in the nerves within various layers of the GI tract and that parasympathetic fibers are disrupted in the gut of individuals with DM [25].

Furthermore, animal studies have demonstrated a decrease in the number of interstitial cells of Cajal (ICCs), the pacemaker cells in the GI tract, essential for the connection between the autonomic nervous system, enteric neurons, and smooth muscle cells [26]. Consequently, the mechanisms of DM-induced changes in sensory-motor function are deeply intertwined with the autonomic nervous system, ENS, and ICCs.

A wide range of factors are related to DAN. The synthesis and the accumulation of advanced glycation end products (AGEs) in peripheral nerves are directly associated with DAN through their effects on structural and functional proteins, and indirectly through the activation of AGE receptors. Additionally, microvascular complications leading to neuropathy involve other biochemical pathways, such as oxidative stress, which is exacerbated by AGE production and polyol pathway activation [27], and the impairment of Na^+^, K^+^-ATPase activity in the cell membrane [28]. Moreover, structural alterations in the nerves, such as endothelial hyperplasia, loss of capillary pericyte coverage, and thickening of the capillary basal membrane, disrupt capillary flow, resulting in a compromise in exchange of glucose and oxygen [29].

Subjects with DM and DAN may experience delayed transit throughout the GI tract, particularly evident in the distal colon [30]. Chandrasekharan et al. demonstrated that colonic musculature in persons with DM does not contract and relax as effectively as that of healthy people, potentially due to the loss of nerve cells in the colon resulting from increased oxidative stress and apoptosis [31].

This prolonged transit is most pronounced in men with long-term T1DM, where the overall colonic transit time is extended [32]. Despite this, the day-night pattern of colonic contractility, characterized by increased contractility from night to morning [33], is maintained in people with T1DM and distal symmetric polyneuropathy [34]. Notably, the gastrocolic reflex appears to be delayed and diminished, particularly in people with concurrent autonomic or peripheral neuropathy [35].

A notable increase in transit time has been observed even in persons with T2DM who do not exhibit neuropathic symptoms. Jorge and colleagues found that the quantity of radiopaque markers in the colon of people with T2DM and healthy controls did not differ significantly 24 h after eating. However, at 72 h post-ingestion, significantly fewer markers were found in healthy subjects [36]. Furthermore, compared to people without constipation, those with DM demonstrated longer colonic transit times [37].

Conversely, animal studies have produced inconclusive results, showing both delayed and accelerated colonic transit in DM, as well as both diminished and enhanced colon contractility. In db/db mice, prolonged transit time is associated with reduced expression of interstitial cells of Cajal (ICCs) and stem cell factor in the colon [38]. Additional contributing factors include a reduction in cholinergic receptors and neuronal nitric oxide synthase in the proximal colon, as well as a downregulation of brain-derived neurotrophic factor (BDNF) and its receptor TrkB [39]. Treatment with insulin-like growth factor 1 (IGF-1) has been shown to prevent apoptosis of colonic smooth muscle cells and improve colon motility in rats with DM [40]. Additionally, treatment with an HRT2B agonist, which acts on the highly selective serotonin type 2B (5-HT_2B_) receptor, enhanced colonic motility in mice with DM [41].

Nonetheless, the loss and damage to ICC in the submucosa and muscle layers are also correlated with increased contractility [42].

Hyperglycemia may significantly contribute to the onset of constipation. Acute hyperglycemia, as opposed to euglycemia, inhibits the colonic contractile response to stomach distension and proximal colonic contraction triggered by colonic distension in healthy individuals [43]. However, in healthy people, acute hyperglycemia has no discernible effects on fasting and post-prandial colonic tone, motility, compliance, and sensations, and on rectal compliance and sensation [44]. In individuals with DM, hyperglycemia may inhibit long and short neural reflexes that influence colonic motility [43].

Regarding chronic hyperglycemia, Ohkuma and colleagues assessed the relationship between defecation frequency and HbA_1c_ in 5131 people with DM from the Fukuoka Diabetes Registry, a multi-center prospective cohort study conducted in a diabetes specialist outpatient clinic [45]. In this cohort (mean age 64.9 years, men 55%), an inverse relationship was observed between defecation frequency and glycemic control. Specifically, subjects with lower defecation frequency (<3 times/week) exhibited higher HbA_1c_ values. This association persisted after multivariable adjustment for confounding factors such as total caloric intake, dietary fiber, body mass index (BMI), treatment with oral antidiabetic medications, and insulin use. Moreover, this relationship did not vary by age or sex. These findings corroborate previous studies [46,47], but, due to the cross-sectional nature of the observation, a cause-and-effect relationship cannot be established without prospective studies. Other limitations include insufficient data on other diabetic complications to determine whether constipation is a consequence of inadequate glucose control or a potential risk factor. As a matter of fact, chronic hyperglycemia may further lead to macroangiopathy and vascular complications, potentially resulting in intestinal ischemia and impairing nerve and colonic muscle function [48]. Additionally, the authors did not address bloating, diarrhea, early satiety, or other GI symptoms [49].

Another significant factor that may contribute to constipation in DM is a high intake of dietary saturated fats. Analysis of data from 6207 adults aged 20 years and older, derived from the 2005–2006 and 2007–2008 cycles of the National Health and Nutrition Examination Surveys (NHANES), revealed a prevalence of constipation of 3.1% in this cohort. After multivariable adjustment, a diet high in saturated fats (exceeding 30 g daily) remained significantly associated with constipation [50]. The odds ratio for high saturated fat intake linked to constipation was notably elevated in subjects with DM over 65 years, particularly among non-Hispanic blacks, females, and those with poor glycemic control, compared to the control group. Several explanations have been proposed for this association: firstly, a high-fat diet may activate the ileal brake, delaying gastric emptying and small bowel transit, thereby increasing the risk of constipation [51,52]; secondly, rats on a high-fat diet exhibit reduced serotonin levels in colon enterochromaffin cells compared to controls [53]; lastly, a high-fat diet may adversely affect human enteric neurons, similar to that observed in mice, through a pathway mediated by microRNA375 upregulation [54].

## 3. Clinical Implications of Constipation in Diabetes Mellitus

In addition to being among the most prevalent GI complaints in DM, constipation represents the most severe symptomatic issue [55]. Clinical manifestations of constipation may include abdominal distension, nausea, vomiting, and electrolyte imbalances [56]. In long-term and more complex scenarios, constipation may also induce stercoral ulcerations and perforation.

A higher frequency of symptoms from the lower GI tract was observed by Kim et al. [57], especially the sensation of incomplete defecation, which was experienced in 62% of subjects, more often than in earlier studies [58,59]. Additionally, individuals with a severe sensation of incomplete defecation experienced anal discomfort, especially evident in people with long-standing DM and DAN symptoms [59]. Given that the anal canal wall and the skin in the anal region are innervated by pudendal nerve fibers [60], discomfort in the anal region could serve as additional evidence of polyneuropathy.

Data regarding gender differences are controversial: Oh et al. [58] and Bytzer et al. [46] reported equal symptom rates in men and women, with no correlation between gender and symptom frequency, respectively, whereas Reszsczynska et al. reported anal region discomfort, sensation of incomplete evacuation, and incontinence significantly more often in women [61].

Furthermore, in a comprehensive prospective cohort study involving 706 individuals with T1DM and 604 control subjects, both groups having a mean age of 41.9 years, it was observed that lower GI symptoms were twice as prevalent among those with T1DM. These symptoms were correlated with diminished QoL and suboptimal glycemic control [62]. Notably, constipation was reported by 8.5% of people with T1DM, compared to 3.3% in the control group, yielding an Odds Ratio (OR) of 2.4. QoL scores were markedly lower in individuals with T1DM and constipation. Except for social functioning and general health perceptions, all QoL scores were significantly impaired in subjects with T1DM, and constipation compared to those with T1DM and regular bowel habits. When comparing individuals with T1DM and constipation to controls with constipation, QoL scores were significantly lower in all domains except social functioning, role limitations due to emotional problems, and mental health composite score. Across all domains, people with T1DM exhibited significantly lower QoL scores than controls with normal bowel habits.

However, unlike other GI symptoms such as diarrhea and abdominal pain, there was no significant difference in HbA_1c_ levels when comparing people with T1DM and constipation to those with T1DM and normal bowel habits, despite a slightly higher value in the first group (8.2% vs. 7.9%, *p* = 0.45) [62].

The same study also examined the association between the frequency of DM complications and any lower GI symptoms. DAN, retinopathy, diabetic kidney disease (DKD), and joint problems were primarily associated with alternating bowel habits and diarrhea, whereas no complications were found to be correlated with constipation alone [62].

Nevertheless, as previously mentioned, constipation and the use of laxatives were independently associated with an increased risk of all-cause mortality and incident CV events, such as coronary heart disease (CHD) and ischemic stroke, in a large cohort of over 3 million US veterans monitored over a 7-year follow-up period [18]. The authors proposed several plausible explanations for this association, including the interactions between alterations in gut microbiota composition and the etiopathogenesis of various illnesses, elevated blood serotonin levels in atherosclerotic disease and constipation, overall autonomic dysfunction, and dehydration resulting from the use of certain medications, particularly diuretics.

Recent cross-sectional studies conducted within the Japanese population have identified a correlation between constipation and certain DM complications. A multicenter study involving 419 people with DM, including 23 with T1DM, demonstrated a significant association between constipation and diabetic retinopathy (OR = 1.99, 95% CI: 1.14–3.45, *p* = 0.015) as well as DAN (OR = 1.86, 95% CI: 1.10–3.16, *p* = 0.021). Furthermore, among subjects with peripheral neuropathy of the lower limbs, the prevalence of constipation was twice as high compared to those without peripheral neuropathy (40.0–49.1% vs. 22.0%) [63]. Data from 410 individuals with T2DM indicated that the prevalence of chronic constipation was 36%, yet only 14% had sought medical consultation for this condition. The prevalence of CHD was more than double in individuals with constipation compared to those without (27% vs. 13%), even after adjusting for age, retinopathy, and DAN. The authors of this study hypothesized that the mechanisms underlying constipation in persons with CHD may involve somatic venous congestion due to increased right atrial pressure and the effects of certain drugs, such as diuretics, beta-blockers, and calcium channel blockers [64]. Additionally, data from the Fukuoka Diabetes Registry revealed that the OR for the presence of DKD in people with T2DM and constipation was 1.58 (95% CI: 1.38–1.82), adjusted for age and sex, compared to those without constipation. This association persisted even after adjustment for other potential confounding factors. Of interest, these findings remained consistent when decreased glomerular filtration rate and urinary albuminuria-to-creatinine ratio were analyzed separately. Potential mechanisms for this association include those previously mentioned: alterations in gut microbiota (leading to the accumulation of uremic toxins), increased serotonin levels, and the use of laxatives [65].

## 4. Constipation as an Adverse Event of Treatment with GLP-1 RAs

Certain antidiabetic medications for T2DM are associated with GI symptoms as adverse events (AEs). Notably, metformin, DPP4 inhibitors, and, particularly, GLP-1 receptor agonists (GLP-1 RAs) have been linked to various GI AEs.

The class of GLP-1 RAs encompasses multiple compounds with distinct pharmacokinetic properties and clinical efficacies, yet they share common mechanisms of action: enhancement of insulin secretion in response to hyperglycemia, suppression of glucagon secretion at hyper- or euglycemia, slowing of gastric emptying to prevent significant postprandial glycemic increases, and lowering caloric intake [66]. Their remarkable impact on glycemic control, without significantly elevating the risk of hypoglycemia, coupled with their efficacy in weight management and maintenance of weight loss, has deeply transformed the management of T2DM and led to their indication in obesity treatment, regardless of T2DM diagnosis. Moreover, certain GLP-1 RAs, such as liraglutide, dulaglutide, and semaglutide in both formulations, have demonstrated CV and renal benefits [66,67,68].

Concerning the reduction in food intake, some findings have indicated the involvement of the arcuate nucleus within the hypothalamus, the area postrema (AP), and the nucleus tractus solitarii (NTS). In this model, GLP-1 RAs seem to be useful in preventing meal initiation by inhibiting the activity of neurons in the arcuate nucleus that produce neuropeptide Y/agouti-related peptide (NPY/AgRP) and causing meal termination in the lateral parabrachial nucleus (PB). Signals reaching the PB originate from the arcuate nucleus of the hypothalamus and brain stem (AP and NTS). POMC/CART neurons that express GLP-1 receptors activate PB neurons and directly or indirectly suppress NPY/AgRP neurons [69,70], leading to disinhibition of suppressive signals to the PB.

The most frequently reported GI side effects of GLP-1 RAs include nausea, vomiting, and diarrhea, although constipation is also commonly observed. These symptoms are typically most pronounced at the initiation of treatment with any GLP1 RA or following dose escalation. Since these symptoms can occur in fasting individuals, they are probably caused by direct interactions with CNS GLP-1 receptors, most likely located in the brain stem (area postrema), rather than the effects of GLP-1 RA treatment on gastrointestinal functions (such as the deceleration of gastric emptying). For most individuals, these episodes are brief and self-limiting, resolving spontaneously with continued treatment [66].

Real-world data remains inconclusive. Data extracted from the US FDA Adverse Event Reporting System (FAERS) database between 2018 and 2022 documented 21.281 reports of GI toxicity. Overall, GLP-1 RAs were associated with increased risk of GI disorders (OR, 1.46; 95% CI, 1.44–1.49) [71]. A total of 1.790 cases of constipation were reported, accounting for 8.41% of total GI events. Notably, these cases were primarily associated with semaglutide therapy, likely due to its more pronounced effects on gastrointestinal motility and neural circuitry compared to other GLP-1 RAs. Conversely, a recent retrospective cross-sectional analysis utilizing real-world data from 10.328 adults with DM or obesity in the National Institutes of Health All of US cohort identified that 3.144 cases of constipation among new users of GLP-1 RAs who reported GI AEs [72]. Furthermore, dulaglutide and liraglutide exhibited higher rates of constipation than semaglutide and exenatide, with a significantly higher rate observed among black people. These discrepancies likely arise from differences in study design, data sources, and populations.

More recently, tirzepatide, the first agonist capable of targeting both GLP-1 and GIP receptors, has been approved for the management of T2DM and obesity. Several studies have confirmed the superiority of tirzepatide in reducing HbA_1c_ levels and body weight compared to GLP-1 RAs and other antidiabetic medications [73]. GIP receptors have also been identified in brain regions involved in the regulation of appetite, satiety, food intake, and energy expenditure, and their agonism produces effects overlapping with those on GLP-1 receptors. However, there appear to be no direct effects on GI motility [74].

The safety profile of tirzepatide was similar to that of GLP-1 RAs, with the same GI AEs reported. Notably, recent data from the FAERS database exhibited a lower incidence of nausea and vomiting compared to GLP-1 RAs [75]. This finding may have a biological basis, given that GIP receptors are found in gamma-aminobutyric acid (GABAergic) neurons of the AP of the brain stem and contribute to the inhibition of neurons responsible for nausea and emesis [76]. However, a significantly higher incidence of constipation was observed, although the possible underlying mechanisms have not yet been investigated [75].

## 5. Diagnosis of Chronic Constipation

The first approach to the diagnosis of constipation is clinical evaluation. A thorough history is needed to determine the cause of constipation, including the patient’s psychological state, obstetric history (for females), and a comprehensive list of all drugs that may be linked to constipation. In order to distinguish between chronic and transient constipation, it is crucial to ascertain the duration of symptoms, the quantity of dietary fiber, and whether the patient has used laxatives or other self-prescribed over-the-counter constipation remedies. Furthermore, before making the diagnosis of chronic constipation, it is critical to screen out malignancy or other organic causes [77].

A complete physical examination is essential, paying particular attention to the abdomen and perineum. Particularly in individuals with DM, a careful rectal examination should be carried out to screen for coexistent pelvic floor muscle dysfunction [78]. The goal of clinical assessment should be to establish a symptom profile in order to design tailored bowel care.

According to evidence-based recommendations, evaluating a patient who suffers from chronic constipation requires a systematic approach. Routine comprehensive diagnostic and physiological tests are not recommended [77,79]. However, physiological tests can be performed to evaluate anorectal disorders in those who are not responding to initial treatment [80,81].

There are various validated techniques for assessing colon transit, including the use of radiopaque markers, colonic transit scintigraphy, and the wireless motility capsule [82]. However, these studies are invasive and usually only available in specialized centers [83]. For this reason, the use of non-invasive methods to assess constipated patients, such as intestinal ultrasound, is currently being evaluated [84].

## 6. Treatment of Chronic Constipation

Treatment of chronic constipation is directed towards alleviating symptoms and regaining intestinal function by accelerating colonic transit and rendering defecation easier [85]. To achieve these clinical objectives, the therapeutic approach can be highly individualized, incorporating both pharmacological and non-pharmacological interventions, often in combination (Table 1).

### 6.1. Dietary and Lifestyle Modifications

Dietary and lifestyle changes can ameliorate symptoms in many individuals and should be implemented before requesting unnecessary diagnostic tests or prescription medications.

A higher-fiber diet, increased water intake, and regular physical activity are integral components of a healthy lifestyle that should be promoted for the patients [86].

Dietary fiber is defined as a plant-derived material resistant to digestion by human enzymes [87]. Fiber is categorized into soluble and insoluble types. Soluble fiber is fermented by colonic bacteria, whereas insoluble fiber is not, playing a crucial role in water retention and promoting a large, bulky stool, thereby enhancing regularity. When consumed in adequate amounts, both soluble and insoluble fibers contribute to improved mineral absorption and alleviation of constipation.

A thorough assessment of dietary fiber and fluid intake is essential before recommending modifications or increases in dietary fiber, as excessive fiber intake can exacerbate gas and bloating and lead to intestinal cramps. Standard recommendations suggest that adults should consume 20–30 g of dietary fiber per day [88]. Consequently, a comprehensive assessment of the type and dosage of dietary fiber is necessary.

Despite controlled trials providing some evidence supporting the efficacy of high-fiber diets, this treatment is ineffective in restoring bowel movements in up to 40% of individuals, particularly those with impaired defecation or slow transit [89,90]. Increased fluid intake is frequently recommended by healthcare professionals for the management of persistent constipation, yet there is limited evidence to substantiate this claim.

Lifestyle modifications are advised as the first line of treatment by the American Diabetes Association (ADA) [91], and these are also the primary interventions for people with persistent constipation. The ADA recommends that most adults with T1DM or T2DM engage in 150 min or more of moderate-intensity aerobic activity per week, spread over at least 3 days a week, with no more than 2 consecutive days without activity. Furthermore, evidence supports encouraging all subjects to reduce sedentary time by briefly standing, walking, or performing other light physical activities at least every 30 min [91]. Given the various lifestyle factors, including changes in routine, irregular exercise, or ignoring the urge to defecate, which can contribute to chronic constipation, lifestyle modifications that incorporate an adequate amount of exercise are typically recommended as an initial management strategy for subjects with DM who exhibit mild to moderate symptoms of chronic constipation.

### 6.2. Pharmacological Management

In instances where dietary and lifestyle modifications prove insufficient, a range of pharmaceutical interventions may be considered.

Laxatives are the most recognized and widely accepted treatment option. However, studies evaluating a progressive approach to laxative therapy are lacking. According to the Rome IV criteria and the Asian Neurogastroenterology and Motility Association, patients who do not respond adequately to bulk-forming agents should first receive treatment with an osmotic laxative, followed by a stimulant laxative [88].

This class of drugs, available in various oral or intrarectal formulations, facilitates bowel movements by softening the stool and/or indirectly stimulating colonic motility through diverse mechanisms [92].

Bulk-forming laxatives, also known as fiber laxatives, enhance stool bulk with soluble fiber, aiding in fluid retention within the stool, thereby making it softer and easier to pass. Commonly prescribed bulk-forming laxatives include *methylcellulose*, *bran*, and *ispaghula/psyllium husk*. Evidence from randomized, placebo-controlled trials (RCTs) indicates that psyllium husk can alleviate symptoms and provide natural relief from constipation by improving stool consistency and colonic transit time [93,94]. It also enhances glycemic control and is safe and well-tolerated in people with DM. Furthermore, it is recognized for its beneficial effects on blood pressure, cholesterol, and body weight [95,96]. One study found that psyllium also reduced the risk of coronary heart disease in subjects with T2DM [97].

Osmotic laxatives function by creating an osmotic gradient that enhances the passage by holding onto more water. Examples of osmotic laxatives include inorganic salts (*magnesium compounds*) and organic sugars or alcohols such as *lactitol*, *lactulose*, and *polyethylene glycol* (*PEG*). They are used to treat both chronic and sporadic constipation, and their dosage can be adjusted according to stool production. Some osmotic laxatives can cause gas, nausea, and bloating in the abdomen [98]. According to the majority of comparative data, lactulose and PEG are equally effective at treating chronic constipation. For example, a randomized, triple-crossover study comparing PEG and lactulose in 57 patients showed that both agents were better than a placebo at improving stool frequency and ease of defecation [99], and similar results were found in another multicenter RCT conducted among 65 patients [100]. However, there is insufficient information on the efficacy and safety of PEG in people with DM, whereas lactulose may have a positive effect on glycemic control because of its fermentation products in the colon, which may disrupt glucose metabolism by reducing glucose hepatic production [101]. There is more limited evidence for the efficacy [102] and safety of magnesium hydroxide; moreover, due to the risk of hypermagnesemia, it should not be used in patients with renal impairment.

Stimulant laxatives, including *bisacodyl*, *senna*, and *sodium picosulfate*, directly stimulate peristalsis and sensory nerve endings, and they are usually used on a short-term basis, with effect commencing within 6 to 12 h [85].

It is crucial to emphasize that both bulk-forming and osmotic laxatives can function as prebiotics, thereby contributing to the resolution of gut dysbiosis and ameliorating inflammatory, metabolic, and molecular imbalances. Gut *microbiota* play a vital role in the digestion and production of organic acids, which are essential for maintaining an appropriate pH environment in the gut. It is well-established that he intestinal microbiota of individuals with DM differs in composition from that of healthy people, characterized by a loss of microbial diversity and an increase in pathogens such as Bacteroidetes and Proteobacteria [103].

Over the years, alternative pharmacological agents have been investigated for the treatment of constipation, particularly in cases where there is an inadequate clinical response to laxatives.

*Prucalopride*, a 5-HT4 receptor agonist, enhances colonic prokinetic activity while exhibiting minimal activity on 5-HT3 and human Ether-à-go-go-Related (hERG) receptors. Three large studies with the same inclusion criteria (laxative-refractory patients, with 80% experiencing less than one spontaneous complete bowel movement per week) showed that prucalopride was linked to an increase in bowel frequency to more than three per week in 24% of patients, compared with 11% in the placebo group [104,105,106].

*Lubiprostone*, a prostaglandin E1 derivate, acts as a chloride-channel activator, increasing intestinal fluid secretion, accelerating colonic transit, and softening stool consistency. Lubiprostone has been shown to be effective in promoting spontaneous bowel movements and relieving self-reported symptoms of chronic constipation [107,108,109].

*Linaclotide*, a guanylate cyclase-C agonist, enhances intestinal fluid secretion and transit. Phase III results indicated that linaclotide improved bowel function in 20% of the patients, with consistent significant improvement in abdominal symptoms, global measures of constipation, and QoL [110]. Common AEs were GI-related, with diarrhea being the most prevalent.

Finally, *elobixibat*, a selective inhibitor of the ileal bile acids transporter (IBAT), has recently been approved in Japan for the treatment of subjects with chronic constipation. This agent is associated with decreased absorption of the bile acids (BAs) and enhanced delivery of BAs to the colon, which subsequently induces colonic contractility, water accumulation, and electrolyte secretion [111]. A single-arm study involving 33 individuals with DM (32 with T2DM, 1 with T1DM) and constipation demonstrated that, after 8 weeks of treatment, elobixibat significantly increased the median frequency of spontaneous bowel movements per week and improved the QoL related to constipation, as assessed by the Japanese version of the Patient Assessment of Constipation Quality of Life, without the occurrence of serious AEs [112]. Another single-arm study, conducted with 21 individuals with T2DM and constipation, revealed a significant positive impact of elobixibat treatment on HbA_1c_ levels and a reduction in LDL cholesterol [113]. These findings corroborate initial studies that demonstrated the benefits of elobixibat on lipid and glucose metabolism, particularly through the elevation of GLP-1 plasma levels [114,115].

It must also be remembered that constipated patients with scarce control of symptoms are often treated (or self-treated) by mechanical methods (enemas) [116].

## 7. Conclusions

Given the increasing global prevalence of DM, healthcare professionals must recognize the clinical problem constituted by the occurrence of chronic constipation. As clinical management of DM advances, facilitating glycemic control, there is potential for the prevention and amelioration of associated GI complications.

Clinical manifestations of constipation can be severe and significantly impact the QoL of individuals with T2DM. Furthermore, it should not be underestimated that in a substantial number of cases, constipation may result from medications commonly prescribed to individuals with DM, including GLP-1 RAs. This can adversely affect patients’ adherence to this class of drugs, posing a significant clinical challenge for clinicians, particularly when constipation occurs in those who mostly need treatment with GLP-1 RAs, such as obese and overweight people with poor glycemic control and who are at elevated CV and renal risk. The development of incretin agonists with reduced impact on GI tolerability may offer a partial solution, although current real-world data available on tirzepatide regarding constipation appear to suggest otherwise.

It is therefore essential to emphasize the importance of a diet rich in fiber and regular physical activity, highlighting the benefits beyond glycemic control, and to communicate this to individuals with DM and concomitant constipation.

Regarding pharmacological treatment, further information is anticipated concerning the potential to address alterations in the intestinal microbiota, which is known to underlie several chronic diseases, as well as the eventual drug interactions between laxatives and other concomitant treatments [117].

Finally, data from intervention studies, particularly head-to-head comparisons, with larger populations and extended durations, are needed to more comprehensively define the clinical potential of the latest drugs approved for the treatment of constipation.

## Figures and Tables

**Figure 1 nutrients-17-03038-f001:**
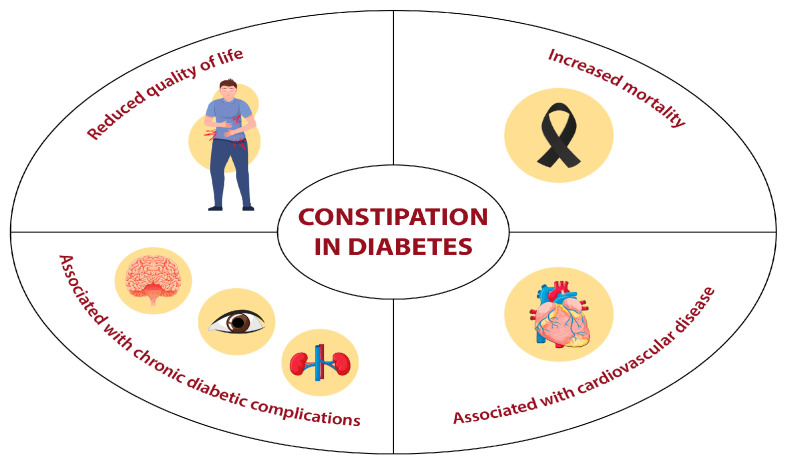
Clinical implications of constipation in diabetes mellitus.

**Figure 2 nutrients-17-03038-f002:**
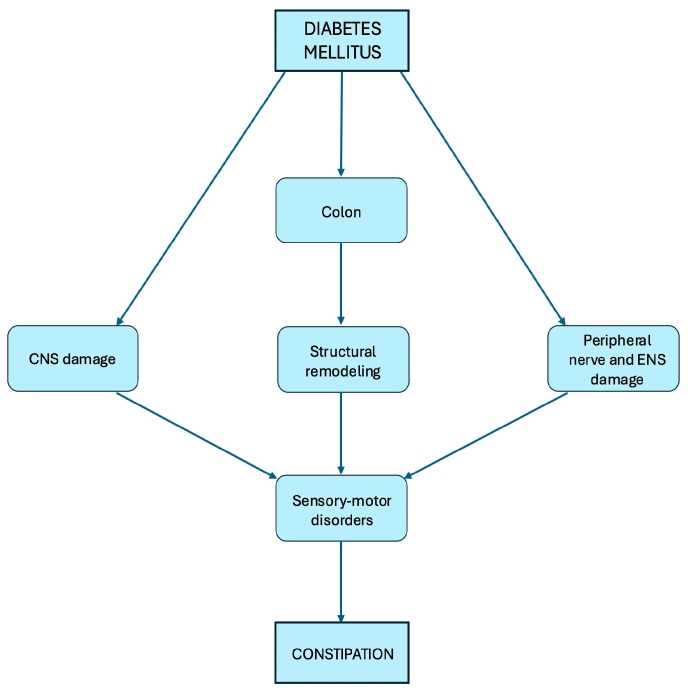
Mechanisms underlying constipation in diabetes mellitus. CNS: Central nervous system; ENS: Enteric nervous system.

**Table 1 nutrients-17-03038-t001:** Therapeutic approach to constipation in persons with diabetes mellitus. 5-HT4: highly selective serotonin type 4; 5-HT3: highly selective serotonin type 3; hERG: human Ether-à-go-go-Related; QoL: Quality of Life; IBAT: ileal bile acids transporter; BAs: bile acids.

Treatment of Constipation				
**Non-pharmacological**				
**Lifestyle**	Diet: More consumption of non-fermentable fibers (consume 20–30 g of dietary fiber per day)		Physical activity: 150 min or more of moderate-intensity aerobic activity per week	
**Pharmacological**				
**Laxatives**	**Category**	**Drugs**	**Mechanism**	**Evidence**
	Bulk-forming	Methylcellulose, bran, ispaghula/psyllium husk	Enhance stool bulk with soluble fiber	First line recommended. better glycemic control.
	Osmotics	Inorganic salts (*magnesium compounds*) and Organic sugars (*lactitol*, *lactulose)*	Generate osmotic gradient with greater water retention in the bowel	Employed for both chronic and occasional constipation
	Stimulant	Bisacodyl, senna, sodium picosulfate	Directly stimulate peristalsis and sensory nerve endings	Only in acute for resistant constipation
**Others**	**Drugs**	**Pharmacokinetics**	**Mechanism**	**Evidence**
	Prucalopride	5-HT4 receptor agonist	Enhances colonic prokinetic activity with activity on 5-HT3 and hERG receptors.	Improvement of constipation in 80% of patients refractory to laxatives
	Lubiprostone	Derivative of prostaglandin E1	Chloride-channel activator: more intestinal fluid secretion, accelerating colonic transit, and softening stool consistency	Improvement of self-reported symptoms of chronic constipation
	Linaclotide	Guanylate cyclase-C agonist	Enhances intestinal fluid secretion and transit	Improvement in abdominal symptoms, global measures of constipation, and QoL in 20% of patients
	Elobixibat	Selective inhibitor of IBAT	enhanced delivery of BAs to the colon, with colonic contractility, water accumulation, and electrolyte secretion	Only in Japan. Improving frequency of spontaneous bowel movements per week

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
