# Peer review of "The Sweet Side of Constipation: Colonic Motor Dysfunction in Diabetes Mellitus"

_nutrients, 2025, doi:10.3390/nu17193038_

Round 1

Reviewer 1 Report

Comments and Suggestions for Authors

Chronic constipation is a common gastrointestinal (GI) disorder in individuals with diabetes mellitus (DM), occurring more frequently than in healthy populations. This review provides a structured overview of this often-underestimated complication in patients with DM. In previous studies, constipation in DM has often been dismissed as a mild symptom, overshadowed by more obvious complications, leading to underdiagnosis and undertreatment. Consequently, this topic is highly relevant and offers a degree of novelty, particularly in exploring the association between GLP-1 receptor agonists (GLP-1 RAs) and chronic constipation in DM. It addresses a significant gap by providing unified data and clinical guidance for managing this underappreciated complication. Notably, the review highlights constipation as an overlooked issue linked to newer diabetes treatments like GLP-1 RAs, which has not been thoroughly examined in earlier literature.
A narrative review of studies up to May 30, 2025, was conducted, encompassing research on the pathogenesis of constipation in DM, its connection to GLP-1 RAs, and diagnostic and therapeutic approaches. This effort effectively fills the void in consolidated information and practical management strategies. The conclusions align well with the presented evidence and arguments, delineating the multifactorial mechanisms of constipation in diabetes, emphasizing the role of GLP-1 RAs, and summarizing current diagnostic and therapeutic strategies. The conclusion aptly underscores the clinical importance of recognizing and addressing this complication.
All references are appropriate and up-to-date, while the tables and figures are well-designed, offering clear and concise summaries of the findings and bibliography. The text is well structured and written. However, the sentences are very long and dense. Phrases such as “is the initial management step” and “a clinical problem arising from the appearance” are garbled and slightly awkward. A more concise wording would strengthen the text. Generally correct, but some issues:“among people with diabetes mellitus (DM), occurring more often than in 13 healthy people” → could be made more specific: “... more often than in healthy people”. Figures and tables provide a clear and concise summary of the findings.” and bibliography up to date. The manuscript would be suitable for publication and of interest to the clinicians ,after careful polishing of the English language.

Comments on the Quality of English Language

The text is well structured and written. However, the sentences are very long and dense. Phrases such as “is the initial management step” and “a clinical problem arising from the appearance” are garbled and slightly awkward. A more concise wording would strengthen the text. Generally correct, but some issues:“among people with diabetes mellitus (DM), occurring more often than in 13 healthy people” → could be made more specific: “...more often than in healthy people”.Figures and tables provide a clear and concise summary of the findings.” and bibliography up to date.The manuscript would be suitable for publication and of interest to the clinicians ,after careful polishing of the English language.

Author Response

Comments 1: [Chronic constipation is a common gastrointestinal (GI) disorder in individuals with diabetes mellitus (DM), occurring more frequently than in healthy populations. This review provides a structured overview of this often-underestimated complication in patients with DM. In previous studies, constipation in DM has often been dismissed as a mild symptom, overshadowed by more obvious complications, leading to underdiagnosis and undertreatment. Consequently, this topic is highly relevant and offers a degree of novelty, particularly in exploring the association between GLP-1 receptor agonists (GLP-1 RAs) and chronic constipation in DM. It addresses a significant gap by providing unified data and clinical guidance for managing this underappreciated complication. Notably, the review highlights constipation as an overlooked issue linked to newer diabetes treatments like GLP-1 RAs, which has not been thoroughly examined in earlier literature.
A narrative review of studies up to May 30, 2025, was conducted, encompassing research on the pathogenesis of constipation in DM, its connection to GLP-1 RAs, and diagnostic and therapeutic approaches. This effort effectively fills the void in consolidated information and practical management strategies. The conclusions align well with the presented evidence and arguments, delineating the multifactorial mechanisms of constipation in diabetes, emphasizing the role of GLP-1 RAs, and summarizing current diagnostic and therapeutic strategies. The conclusion aptly underscores the clinical importance of recognizing and addressing this complication. All references are appropriate and up-to-date, while the tables and figures are well-designed, offering clear and concise summaries of the findings and bibliography. The text is well structured and written.]

Response 1: [We thank the Reviewer for his/her appreciation regarding the content and style of our manuscript. We are really proud of this.]

Comments 2:  [However, the sentences are very long and dense. Phrases such as “is the initial management step” and “a clinical problem arising from the appearance” are garbled and slightly awkward. A more concise wording would strengthen the text. Generally correct, but some issues:“among people with diabetes mellitus (DM), occurring more often than in 13 healthy people” → could be made more specific: “... more often than in healthy people”. Figures and tables provide a clear and concise summary of the findings.” and bibliography up to date. The manuscript would be suitable for publication and of interest to the clinicians ,after careful polishing of the English language.]

Response 2: [We thank the Reviewer for his/her suggestions. Accordingly, we have revised the manuscript, trying to improve the English language and eliminate the phrases highlighted by the Reviewer. Surely, these suggestions have improved the quality of our article.]

Reviewer 2 Report

Comments and Suggestions for Authors

This review article provides a clear and detailed overview of the possible causes of colonic motor dysfunction (i.e. constipation) as observed to occur in diabetic patients. The most common pathophysiological causes leading to constipation are mentioned and properly referenced, and the most common modalities of assessment and treatment are reported clearly. The concepts are clearly stated and the conclusions are in line with the information provided in the review. The abstract is to the point and states clearly the purpose of the review and its main findings

Comments on the Quality of English Language

English style and syntax are fine. Concepts are clearly expressed

Author Response

Comment: [This review article provides a clear and detailed overview of the possible causes of colonic motor dysfunction (i.e. constipation) as observed to occur in diabetic patients. The most common pathophysiological causes leading to constipation are mentioned and properly referenced, and the most common modalities of assessment and treatment are reported clearly. The concepts are clearly stated and the conclusions are in line with the information provided in the review. The abstract is to the point and states clearly the purpose of the review and its main findings]

Response: [We thank the Reviewer for the many compliments on our article. We're really proud that both the content and the style were appreciated. Additionally, we have improved the English language according to the valuable suggestion of the Reviewer.]